# Integrative bioinformatics and machine learning identify shared molecular mechanisms and diagnostic biomarkers between *Helicobacter pylori* infection and atrial fibrillation

Aojian Deng[1], Wei Wang[ID][2]*

**1** Department of Gastroenterology, The Third Xiangya Hospital, Central South University, Changsha, Hunan, China, **2** Department of Fourth Internal Medicine, The Third People's Hospital Health Care Group of Cixi, Ningbo, China

* WangWei971122@163.com

## Abstract

### Background

*Helicobacter pylori* (*H. pylori*) infection and atrial fibrillation(AF) are major global health concerns. Emerging evidence has suggested a potentially chronic inflammation-mediated link between them, but the shared genetic mechanisms remain unclear.

### Methods

We analyzed multidataset gene expression profiles from the Gene Expression Omnibus (GEO) database. Differential expression analysis, weighted gene co-expression network analysis (WGCNA), functional enrichment, and machine learning were employed to identify common genes, pathways, and diagnostic biomarkers. Protein-protein interaction (PPI) networks, drug-gene analysis, and molecular docking were used to identify hub genes and potential therapeutics.

### Results

We identified 73 common differentially expressed genes (DEGs) between *H. pylori* infection and AF, which were predominantly enriched in immune-related processes including leukocyte activation, neutrophil migration, and myeloid cell-mediated immunity. Machine learning identified 15 and 23 key feature genes for *H. pylori* and AF, respectively, with S100A8 emerging as a shared diagnostic biomarker. Ten hub genes including TYROBP, ITGB2, and SPI1, were identified from the PPI network. Drug repositioning analysis suggested retinoic acid, indirubin, and ropivacaine as candidate therapeutics targeting these key hub genes.

**Data availability statement:** All sequencing data were sourced from the Gene Expression Omnibus (GEO) database. A total of four datasets related to AF were included: GSE41177 and GSE176166 constituted the discovery cohort, while GSE108660 and GSE128188 formed the validation cohort. For H. pylori infection, five datasets were selected: GSE27411, GSE60662, and GSE233973 comprised the discovery cohort, and GSE5081 and GSE60427 served as the validation cohort.

**Funding:** The author(s) received no specific funding for this work.

**Competing interests:** The authors have declared that no competing interests exists.

## Conclusion

Our integrative analysis highlights the central role of immune-inflammatory pathways in linking *H. pylori* infection to AF. We propose S100A8 and other identified hub genes as potential biomarkers and therapeutic targets. The predicted candidate therapeutics, particularly retinoic acid, may offer novel avenues for intervention, warranting further experimental validation.

## Introduction

*Helicobacter pylori* (*H. pylori*) is a gram-negative, microaerophilic, spiral-shaped bacterium that colonizes the human gastric mucosa and represents one of the most prevalent chronic bacterial infections worldwide [1,2]. It is estimated that more than half of the global population is infected, with acquisition typically occurring during childhood. Without treatment, the infection often persists through life [2,3]. *H. pylori* is a primary etiological agent of chronic active gastritis and is strongly associated with peptic ulcer disease—which causes approximately 90% of duodenal ulcers and 70–80% of gastric ulcers—as well as gastric mucosa-associated lymphoid tissue (MALT) lymphoma [4].

Beyond its direct gastrointestinal manifestations, accumulating evidence hsa indicated potential links between *H. pylori* infection and various extragastric disorders, including unexplained iron-deficiency anemia, idiopathic thrombocytopenic purpura, and cardiometabolic diseases. This broad influence positions *H. pylori* as a systemic pathogen capable of affecting multiple organ systems [5].

Atrial fibrillation (AF) is the most prevalent sustained cardiac arrhythmia and id characterized by rapid and disorganized atrial electrical activity resulting in ineffective atrial contraction. Clinical presentation varies widely; patients may experience palpitations, fatigue, and dyspnea, or remain entirely asymptomatic [6,7]. The global prevalence of AF is steadily increasing, affecting tens of millions of adults and rising significantly with age, imposing a substantial and growing public health burden [8].

The clinical significance of AF extends beyond its symptomatic burden. Its most critical complication is intracardiac thrombus formation secondary to atrial blood stasis; subsequent embolization can lead to systemic embolism, particularly ischemic stroke, increasing the risk of stroke five-fold [9,10]. Additionally, AF is independently associated with heart failure, cognitive impairment, and elevated all-cause mortality [11–13].

The pathophysiology of AF is multifactorial. The established risk factors include advanced age, hypertension, heart failure, valvular heart disease, diabetes, obesity, and sleep apnea. The initiation and perpetuation of AF are driven primarily by electrophysiological and structural remodeling processes, including atrial fibrosis, ion channel dysfunction, inflammation, and oxidative stress [13]. Notably, systemic inflammation has been recognized as an independent contributor to AF pathogenesis. Inflammatory mediators can promote atrial fibrosis and alter electrophysiological properties, thereby creating a substrate conducive to AF initiation and maintenance [14].

Emerging clinical evidence has suggested a potential association between *H. pylori* infection and an increased risk of AF, implicating that it is involved in the development of arrhythmia. However, the specific underlying mechanisms remain insufficiently elucidated [15–17]. In this study, we addressed this gap by leveraging public sequencing data for *H. pylori* and AF from the GEO database. Through integrated bioinformatic analyses, machine learning approaches, and molecular prediction methods, we aimed to identify shared hub genes, elucidate common pathways, and explore potential therapeutic molecules associating these two prevalent conditions.

## Materials and methods

### Data acquisition and preprocessing

All gene expression data were sourced from the GEO database. Four datasets were included for AF: GSE41177 and GSE176166 constituted the discovery cohort, while GSE108660 and GSE128188 formed the independent validation cohort [18,19]. Five datasets were selected for *H. pylori* infection: GSE27411, GSE60662, and GSE233973 comprised the discovery cohort, and GSE5081 and GSE60427 serving as the validation cohort [20–24]. Detailed information for each dataset, including platform, sample sizes (disease vs. control), and source, is summarized in Table 1. Raw data (CEL files) were processed using the affy or oligo R packages for background correction and normalization. For the discovery cohorts within each condition (H. pylori infection and AF), the sva R package was used to perform ComBat batch effect correction, integrating multiple datasets into a single expression matrix for subsequent differential expression and WGCNA analyses [25]. Validation cohorts were processed individually without batch correction to ensure independent assessment.

### Screening and functional enrichment of DEGs

DEGs between disease and control samples in the discovery cohorts were identified separately for *H. pylori* infection and AF using the limma R package. Datasets related to AF and *H. pylori* infection were analyzed separately to identify both up- and down-regulated DEGs for each condition [24]. Genes with an adjusted p-value (adj.P.Val) < 0.05 and |log2 fold change (FC)| > 0.5 were considered significant. Commonly co-expressed DEGs between AF and *H. pylori* infection were extracted on the basis of their overlapping expression patterns (shared up- or down-regulation). Functional enrichment analysis of these common DEGs was performed using the org.Hs.e.g.,db and clusterProfiler R package for Gene Ontology (GO) terms and Kyoto Encyclopedia of Genes and Genomes (KEGG) pathways.

### WGCNA

WGCNA was employed on the batch-corrected discovery cohort expression matrices to construct scale-free co-expression networks. The soft-thresholding power was chosen based of the criterion of approximate scale-free topology. A topological

**Table 1. Basic information of the GEO datasets used in the study.**

| GSE series | Disease | Samples | Platform | Group |
|---|---|---|---|---|
| GSE41177 | AF | 32 AF samples and 6 healthy controls | GPL570 | Discovery cohort |
| GSE176166 | AF | 3 AF samples and 4 health controls | GPL23126 | Discovery cohort |
| GSE108660 | AF | 5 AF samples and 5 healthy controls | GPL19612 | Validation cohort |
| GSE128188 | AF | 5 AF samples and 5 healthy controls | GPL18573 | Validation cohort |
| GSE27411 | Hp | 6 Hp samples and 6 healthy controls | GPL6255 | Discovery cohort |
| GSE60662 | Hp | 12 Hp samples and 4 healthy controls | GPL13497 | Discovery cohort |
| GSE233973 | Hp | 13 Hp samples and 9 healthy controls | GPL21185 | Discovery cohort |
| GSE5081 | Hp | 16 Hp samples and 16 healthy controls | GPL570 | Validation cohort |
| GSE60427 | Hp | 24 Hp samples and 8 healthy controls | GPL17077 | Validation cohort |

overlap matrix (TOM) was constructed, and genes were clustered into modules using dynamic tree cutting. Module eigengenes (MEs) were computed, and their correlations with the clinical trait (disease status) were assessed. Modules with the highest absolute correlation ($|r| > 0.4$, $p < 0.05$) were selected as trait-relevant. Genes with high module membership (MM > 0.8) and gene significance (GS > 0.2) within these key modules were considered core genes.

### Application of multiple machine-learning (ML) methods

Data Preparation and Modeling Strategy: The union set of genes from common Differentially expressed genes DEGs and shared WGCNA module genes served as the initial feature space. For each condition (*H. pylori* and AF), the batch-corrected discovery cohort was used for model training and feature selection. We employed more than 20 ML algorithms, including Random Forest (RF), least absolute shrinkage and selection operator (LASSO) regression, and eXtreme Gradient Boosting (XGBoost), which were implemented using R packages such as glmnet, randomForest, and XGBoost, respectively. A 10-fold cross-validation (CV) repeated 5 times within the discovery cohort was used for hyperparameter tuning and to prevent overfitting. The optimal hyperparameters (e.g., mtry for RF, lambda for LASSO) were selected on the basis of the highest mean AUC from the CV process. Each algorithm generated a ranked list of important features. Gene Selection and Integrated Model Building: The top-performing features from each single algorithm were intersected to derive a consensus set of pivotal diagnostic genes. An integrated model was then constructed using a two-step approach: first, a meta-classifier (either naïve Bayes for *H. pylori* or linear discriminant analysis (LDA) for AF) was trained on the predictions of the base RF model as new input features. The diagnostic performance of the final model was evaluated in the independent validation cohorts using the area under the receiver operating characteristic curve (AUROC). The contribution of each selected gene to the RF model's prediction was quantified using SHapley Additive exPlanations (SHAP) analysis.

### PPI network construction and hub gene identification

A PPI network for the co-expressed DEGs between AF and *H. pylori* as input was constructed using the STRING database (v12.0) with a confidence score threshold > 0.4. Hub genes were identified using the CytoHubba plugin, applying the maximal clique centrality (MCC) and degree algorithms. The top 10 overlapping candidates from both algorithms were selected as hub genes. The functional interactions among these hub genes were further explored using the GeneMANIA web tool [25].

### Screening and structural characterization of small molecules

The identified hub genes were submitted to the DSigDB database of the ENRICHR platform to predict potential therapeutic compounds. Compounds with an adjusted p-value < 0.05 were considered significant [26–28]. The chemical structures of the top candidates were retrieved from PubChem. Molecular docking was performed to evaluate the binding affinity between the candidate drugs and their predicted target proteins (hub genes). Protein structures were obtained from the RCSB Protein Data Bank (PDB). If an experimental structure was unavailable for a human target, homology modeling was performed using the SWISS-MODEL server. Docking simulations were conducted using the CB-Dock2 web server, which predicts binding sites and calculates Vina scores. The conformation with the lowest (most negative) Vina score for each pair was selected for analysis and visualization using PyMOL.

## Results

A schematic workflow of the present study was depicted in Fig 1, which systematically summarized the entire analytical pipeline from raw data acquisition to the final model performance evaluation. Briefly, the study commenced with the collection and preprocessing of original multi-batch omics data, where batch effects were effectively eliminated using the ComBat batch correction method (S2 Fig) to ensure the reliability and comparability of subsequent analyses. After

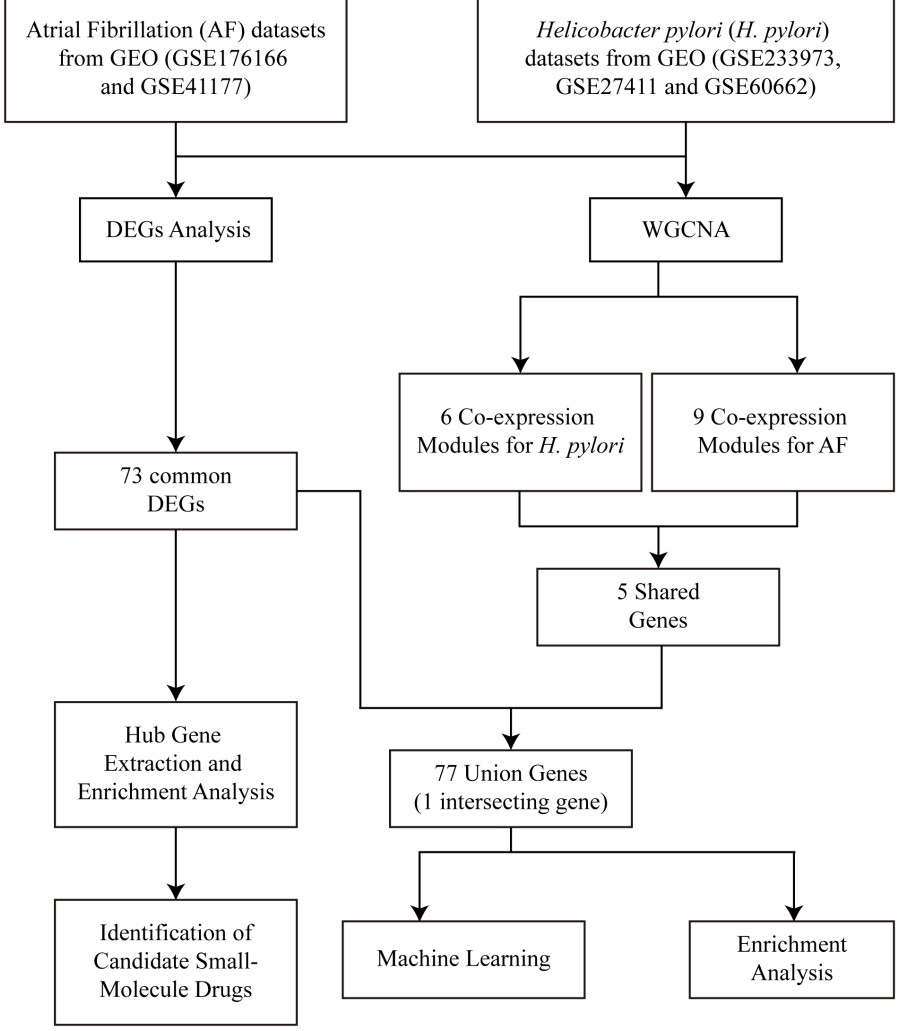

**Fig 1. Schematic Workflow of the Research.**

data preprocessing, key feature screening and dimensionality reduction were conducted to extract the most informative variables for model construction. Subsequently, the binary classification model was established based on the optimized feature set, and the Matthews Correlation Coefficient (MCC Score) was employed as the core metric to comprehensively evaluate the predictive performance of the model. Finally, the robustness and generalization ability of the model were further verified through cross-validation and independent external validation, thus completing the whole research process of data analysis, model building and performance assessment.

## Identification of common DEGs in *H. pylori* infection and AF

To establish the foundational transcriptional overlap between H. pylori infection and AF, we first performed differential expression analysis on their respective discovery cohorts. Regarding H. pylori infection, we identified 2,160 DEGs (1,334 upregulated and 826 downregulated). In terms of AF, 414 DEGs were identified (210 up- and 204 down-regulated) (Fig 2A–D). Functional enrichment revealed that DEGs related to both conditions were significantly associated with immune

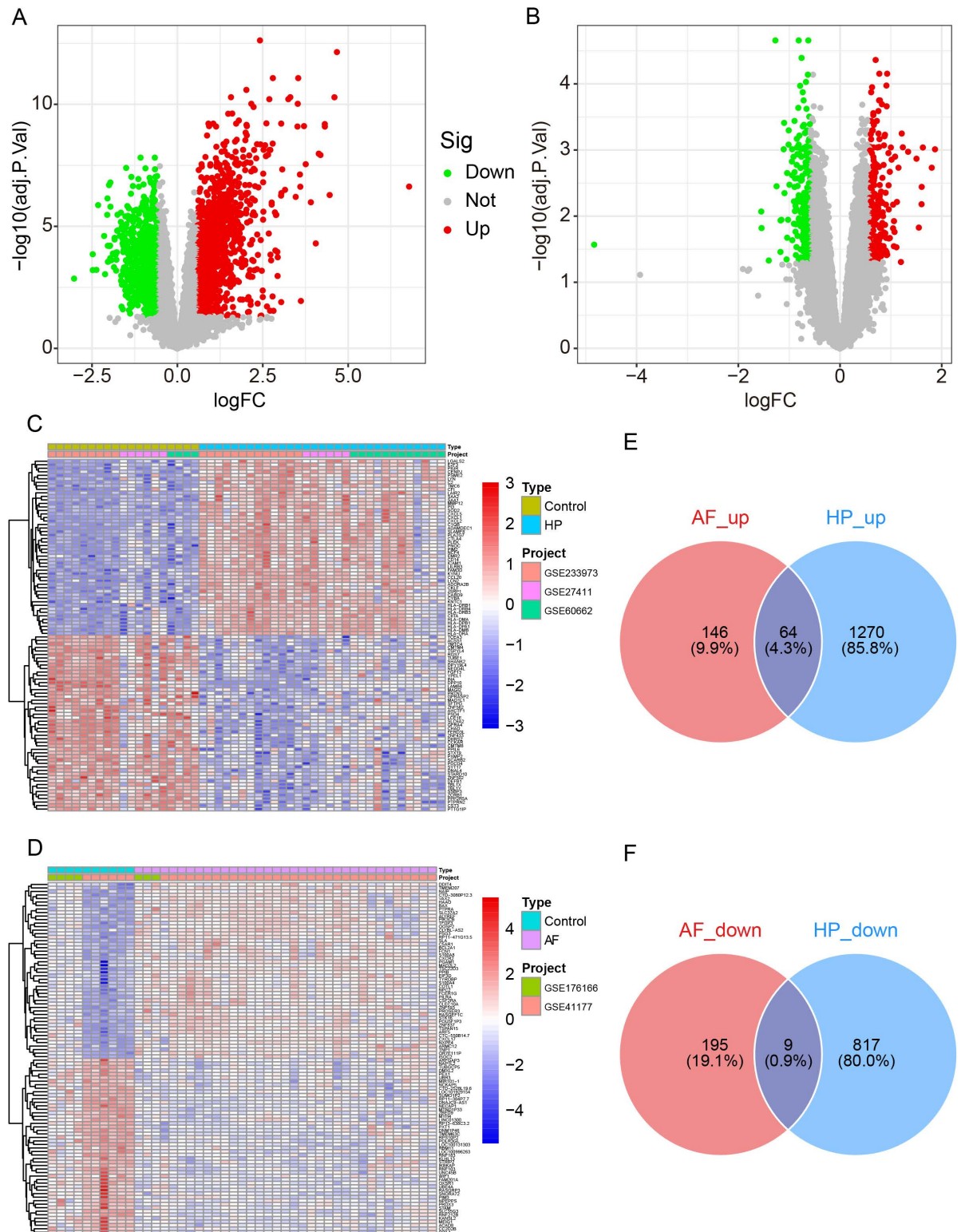

**Fig 2. Identification of common DEGs in *H. pylori* infection and AF.** (A–B) Volcano plots of all DEGs in the *H. pylori* and AF discovery cohorts. Red dots represent upregulated genes, green dots represent downregulated genes, and gray dots represent genes with no differential expression (adjusted p value < 0.05, |log2FC| > 0.5). (C–D) Heatmaps of the top 50 DEGs in the *H. pylori* and AF discovery cohort. (E–F) Venn diagram showing 64 coupregulated and 9 codownregulated DEGs.

cell regulation and proliferation (S1 Fig). Critically, an intersection of these gene sets revealed 73 common DEGs shared between the two diseases, comprising 64 consistently upregulated genes and 9 downregulated DEGs (Fig 2E, F). Functional enrichment analysis indicated that the DEGs for H. pylori were primarily associated with immune cell proliferation, differentiation, and regulation. Similarly, DEGs for AF were enriched in immune cell-related pathways, as well as those implicated in hematologic disorders (S1 Fig). This core gene set provided the initial evidence for a shared transcriptional response.

## WGCNA identifies trait-relevant gene modules

To move beyond individual genes and identify functionally coherent gene networks associated with each condition, we employed WGCNA. In the H. pylori cohort (soft threshold $\beta = 20$), the black ($r = 0.77$, $p = 1 \times 10^{-10}$), brown ($r = 0.68$, $p = 5 \times 10^{-8}$), and gray ($r = 0.52$, $p = 1 \times 10^{-4}$) modules strongly positively correlated with infection status, whereas the yellow module ($r = -0.68$, $p = 5 \times 10^{-8}$) was negatively correlated (Fig 3A–C). In the AF cohort (soft threshold $\beta = 5$), the brown ($r = 0.45$, $p = 0.002$) module was positively correlated, whereas the black ($r = -0.57$, $p = 5 \times 10^{-5}$) and magenta ($r = -0.41$, $p = 0.005$) modules were negatively correlated with the disease trait (Fig 3D–F). The intersection of genes from these key, trait-relevant modules across both diseases yielded an additional 5 shared genes (Fig 3G). This network-based approach independently reinforced the existence of a common genetic substrate.

## Functional enrichment of the integrated gene set implicates immune-inflammatory pathways

We subsequently merged the 73 common DEGs with the 5 shared WGCNA genes into a unified set of 77 unique genes (including 1 intersecting gene) for comprehensive functional annotation. GO enrichment analysis revealed that these genes are overwhelmingly involved in immune effector processes, including leukocyte-mediated immunity, leukocyte proliferation, myeloid leukocyte activation, and neutrophil chemotaxis/migration (Fig 4B, C). KEGG pathway analysis further highlighted the roles of these genes in innate immune pathways, such as the complement and coagulation cascades, hematopoietic cell lineage, cell adhesion molecules, natural killer cell-mediated cytotoxicity and neutrophil extracellular trap formation (Fig 4D, E). These results strongly suggest that dysregulated immune-inflammatory signaling forms a central biological link between these two conditions.

## ML identifies robust diagnostic biomarkers

To distill the shared gene set into a minimal, high-fidelity diagnostic signature, we conducted a comprehensive comparison of more than 20 ML algorithms. The feature selection process for *H. pylori* infection and AF is detailed in Fig 5A and 5F. For *H. pylori* infection, an integrated random forest + naïve model identified 15 pivotal genes, with SHAP analysis ranking S100A9 (SHAP = 0.359), S100A8 (0.308), C1QA (0.261), HLA-DPA1 (0.211), and CD74 (0.181) as the top contributors (Fig 5B). This model demonstrated excellent diagnostic performance (AUROC = 0.93) in the independent validation cohort (Fig 5E). For AF, an optimal random forest + LDA model selected 23 pivotal genes, with PGAM1 (SHAP = 0.0237), S100A8 (0.0202), SLA (0.0195), C5AR1 (0.018), and CD28 (0.017) being the most informative features according to the SHAP values (Fig 5G). This finding was also robust (AUROC = 0.88) (Fig 5J). The differential expression patterns of these key genes are summarized in Fig 5D and 5I. The consistent appearance of S100A8 as a key feature in both disease-specific models underscores its potential as a shared diagnostic biomarker.

## Hub gene extraction and enrichment analysis

To elucidate the core regulatory machinery within the shared gene set, we constructed a PPI network. CytoHubba analysis revealed the top 10 hub genes, including TYROBP, ITGB2, ITGAM, and SPI1 (Fig 6B and Table 2). Functional analysis of these hub genes confirmed and refined our earlier findings, showing concentrated enrichment in myeloid leukocyte activation and neutrophil degranulation (Fig 6D–F). KEGG pathways such as neutrophil extracellular trap formation and natural

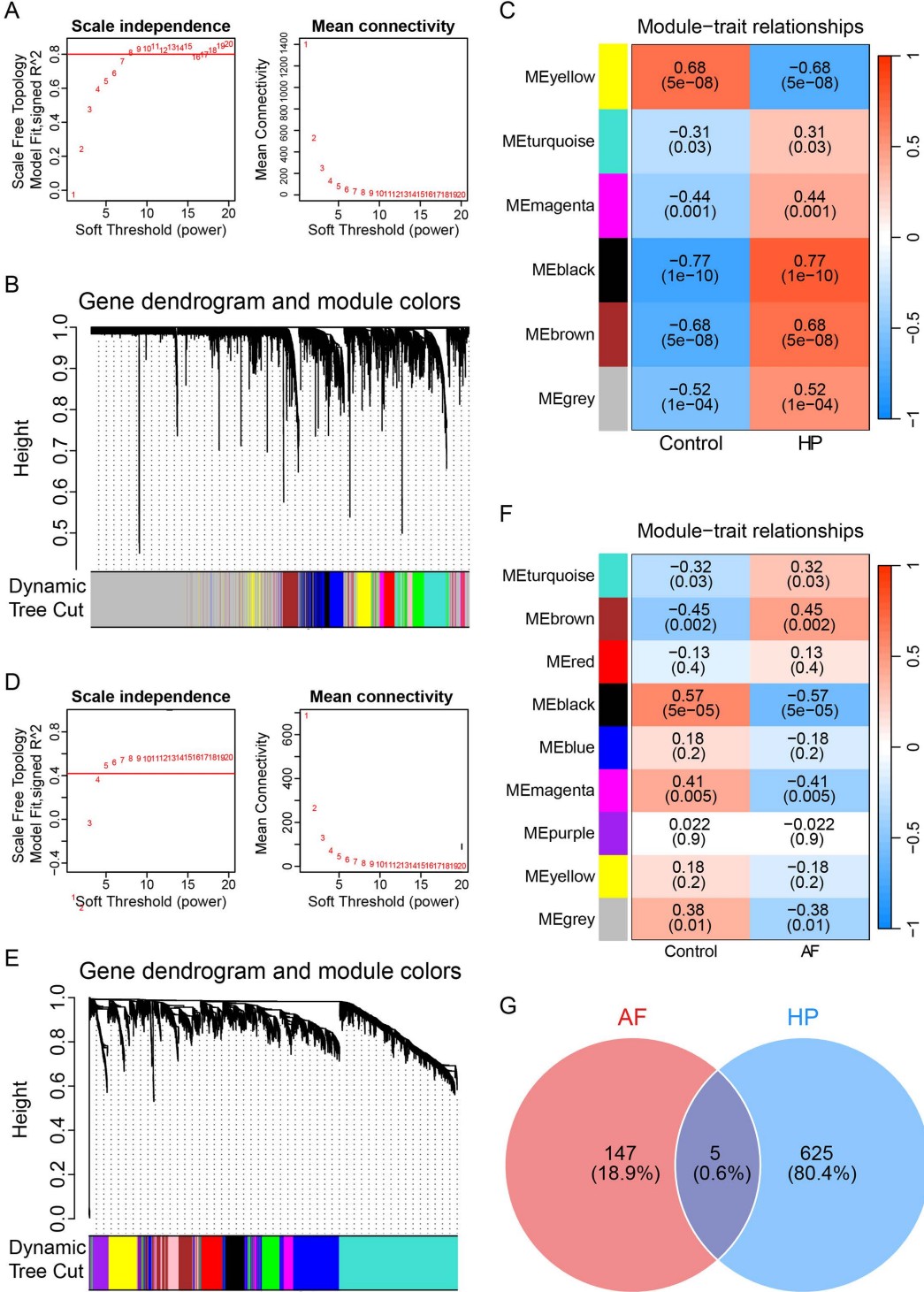

**Fig 3. WGCNA identifies trait-relevant gene modules.** Analysis of network topology for various soft-thresholding powers (β) to select the optimal value for constructing a scale-free network in the *H. pylori* infection (A) and AF (D) cohorts. Hierarchical clustering dendrograms of genes for *H. pylori* infection (B) and AF (E) by the dynamic tree cut algorithm. Heatmaps of the module–trait relationships showing the association (Pearson's r) between module eigengenes and clinical traits in *H. pylori* infection (C) and AF (F). Each cell contains the correlation coefficient and corresponding p-value. The darker the color is, the stronger the correlation, red indicates a positive correlation, and blue indicates a negative correlation. (G) Venn diagram demonstrating the intersection of common genes obtained by WGCNA from the key disease-correlated modules identified in both *H. pylori* infection and AF analyses, yielding 5 shared genes.

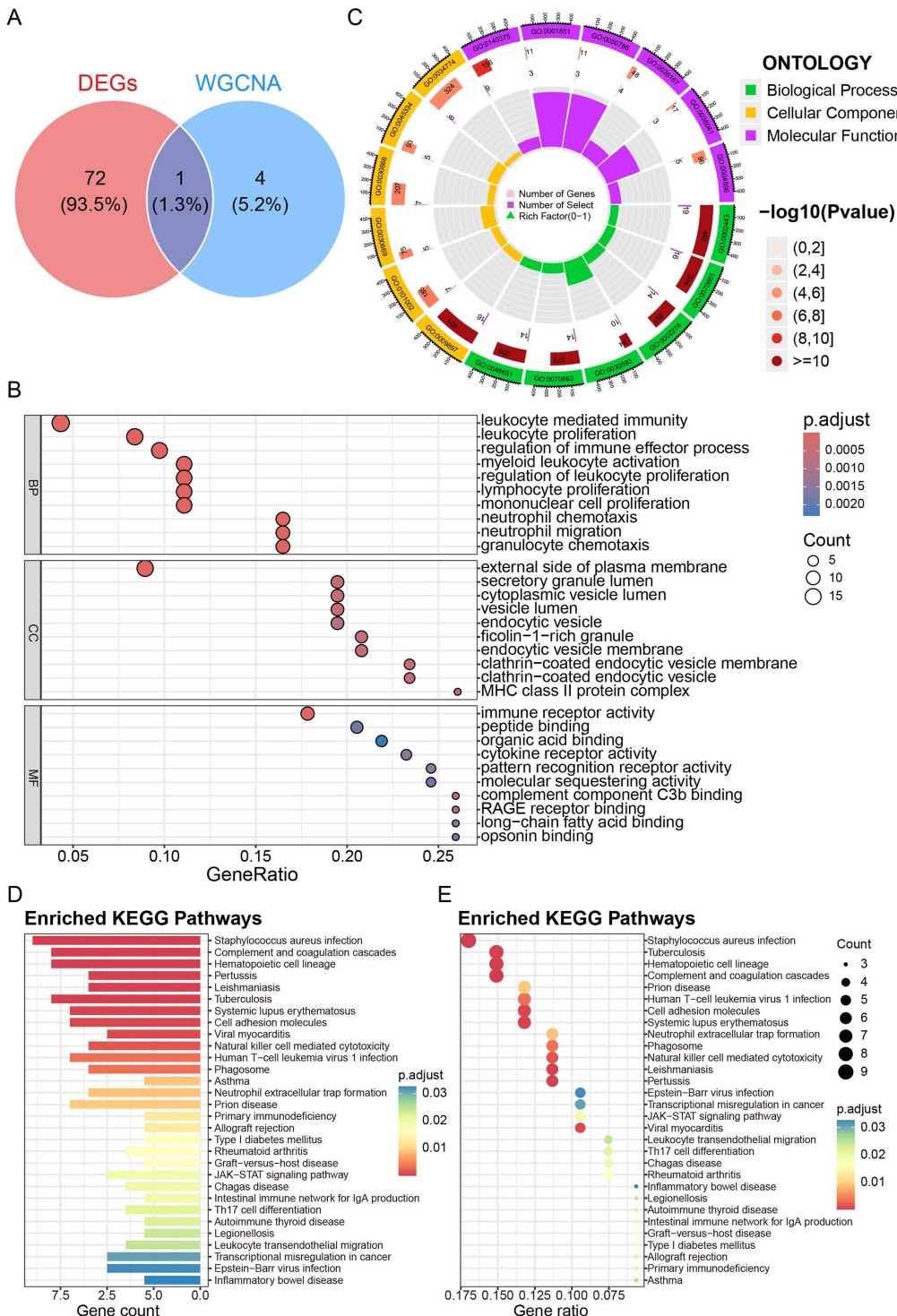

**Fig 4. Functional enrichment of the integrated gene set implicates immune–inflammatory pathways.** (A) Venn diagram visualizing the composition of the 77-gene set derived from common DEGs and shared WGCNA genes. (B–C) Results of the enrichment analysis of the enriched genes in the GO pathway. Dot plot (B) and circular plot (C) of the top significantly enriched GO terms. BP, biological process; CC, cellular component; MF, molecular function. (D–E) Bar plot (D) and dot plot (E) displaying the KEGG pathway enrichment analysis results. The color scale represents the adjusted p value, and the dot size corresponds to the gene count.

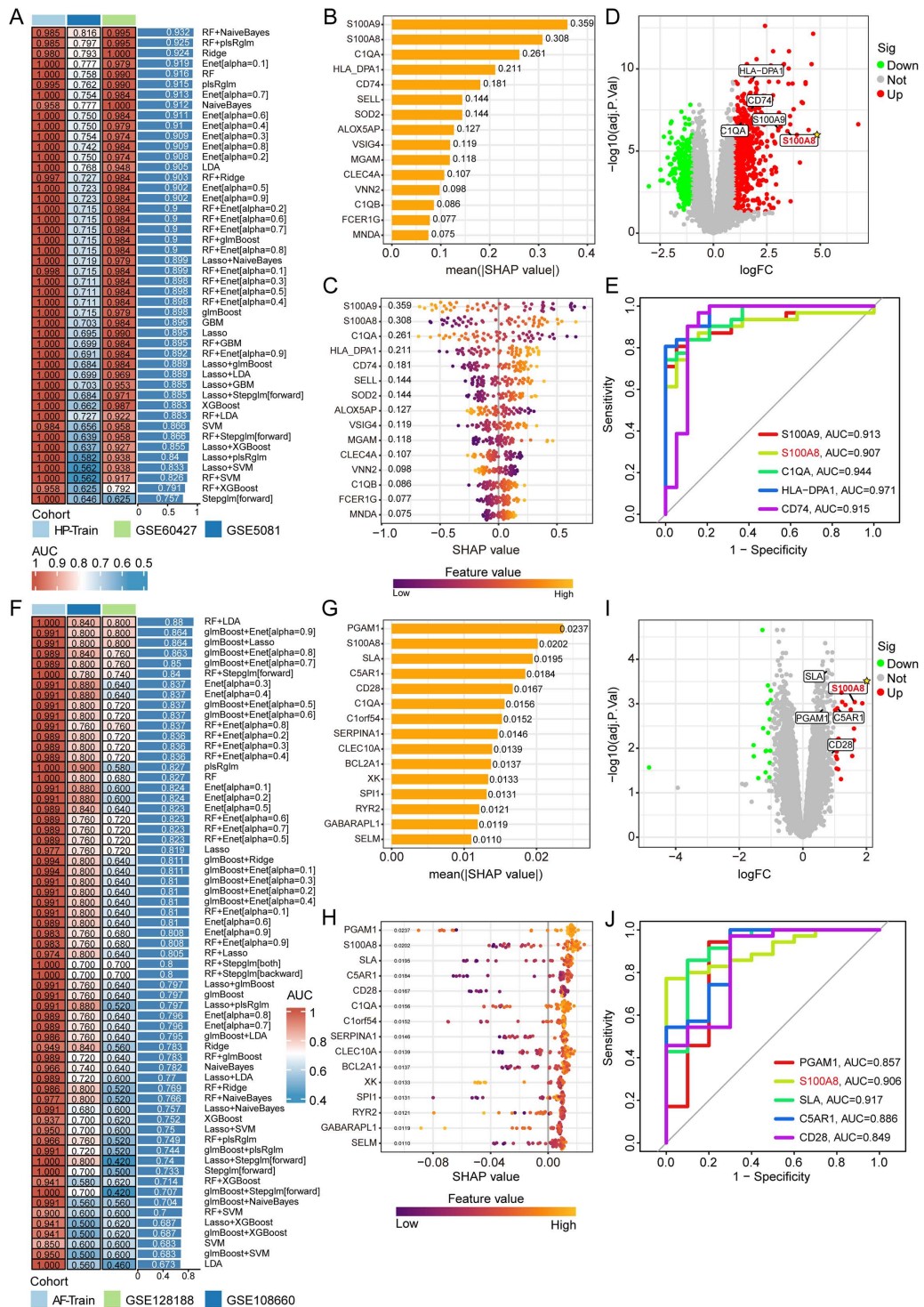

**Fig 5. ML identifies robust diagnostic biomarkers.** (A, F) Model performance comparison: Heatmap showing AUC values for various models across cohorts in *H. pylori* infection (A) and AF (F). Right column: models; middle column: AUC. Colors indicate cohort sources. (B, G) Bar plot of gene SHAP values incorporated in the optimal ML methods for *H. pylori* infection (B) and AF (G). Features are ordered by their mean absolute SHAP value. The larger the bar is, the greater the contribution to the model. (C, H) Violin plots showing gene expression distributions across conditions in *H. pylori* infection (C) and AF (H). Width represents the density of the data, and color represents the level of expression. (D, I) Volcano plot of the top 5 genes incorporated in the optimal ML methods for *H. pylori* infection (D) and AF (I). (E, J) ROC curve of the top 5 genes incorporated in the optimal ML methods for *H. pylori* infection (E) and AF (J). The AUC values are displayed.

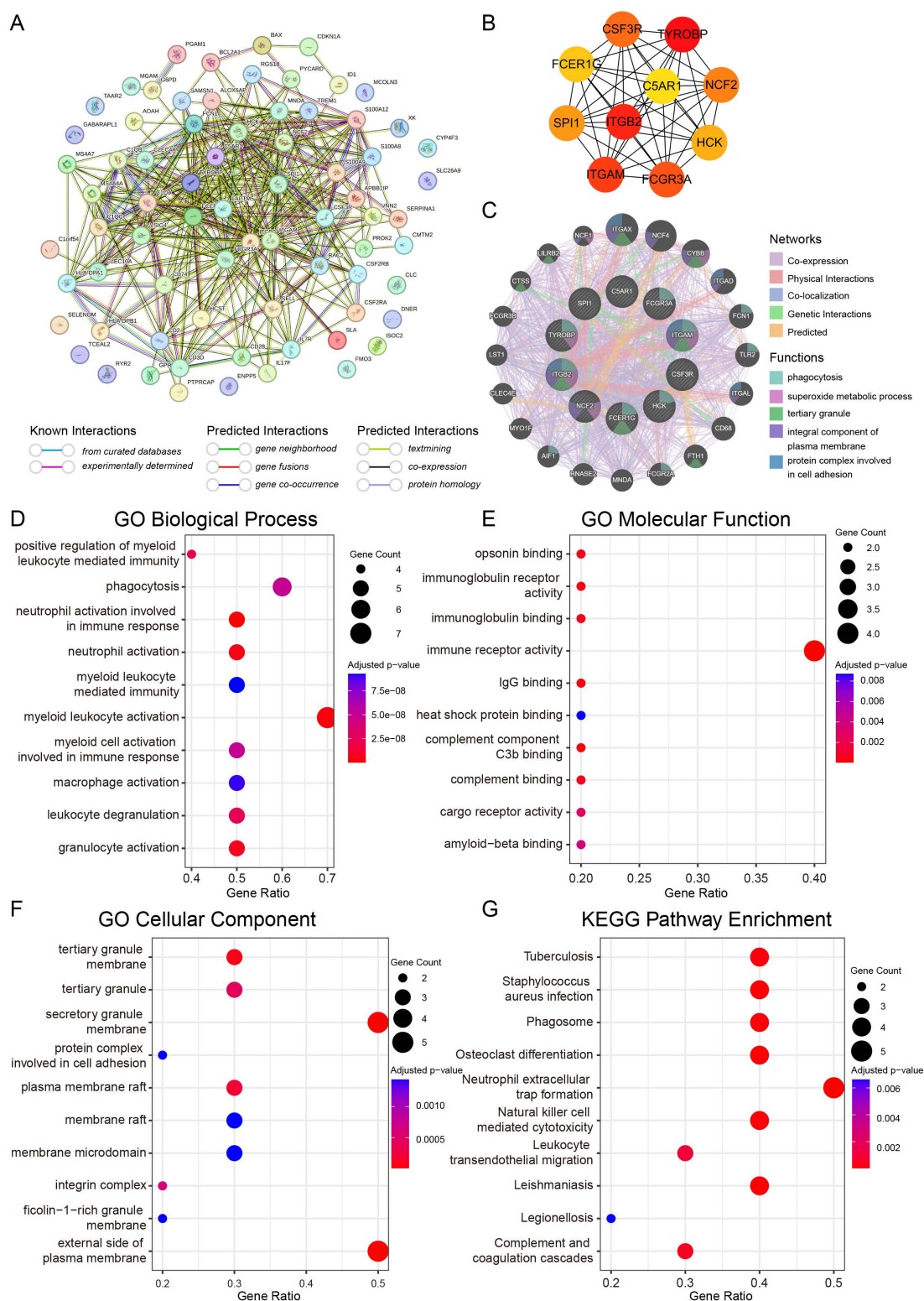

**Fig 6. Hub gene extraction and enrichment analysis.** (A) PPI network analysis of DEGs (STRING database). (B) The top 10 hub genes identified by the degree and the maximal clique centrality (MCC) algorithms via the CytoHubba plugin using the MCC method in Cytoscape. (C) GeneMANIA diagram showing the coexpression interactions between the hub genes and their neighboring genes. The color codes indicate the functions shared by the genes. (D–F) GO enrichment analysis of the hub genes, showing the top terms for BP (D), MF (E), and CC (F). (G) KEGG enrichment analyses of the hub genes.

**Table 2. Top 10 hub genes of the shared gene set.**

| Rank | Name | MCC Score |
|---|---|---|
| 1 | TYROBP | 2.05E+07 |
| 2 | ITGB2 | 2.05E+07 |
| 3 | ITGAM | 1.99E+07 |
| 4 | FCGR3A | 1.98E+07 |
| 5 | CSF3R | 1.42E+07 |
| 6 | NCF2 | 1.39E+07 |
| 7 | SPI1 | 1.38E+07 |
| 8 | HCK | 1.35E+07 |
| 9 | FCER1G | 1.10E+07 |
| 10 | C5AR1 | 9999366 |

MCC Score: Matthews Correlation Coefficient, a comprehensive performance metric for binary classification models, with a value range of [−1, 1]. A higher value indicates better model prediction performance.

killer cell-mediated cytotoxicity were again prominent (Fig 6G), indicating that these hub genes occupy central positions in the inflammatory networks linking *H. pylori* infection to AF.

## Prediction and validation of potential therapeutic compounds

Finally, to explore translational implications, we leveraged the identified top 10 hub genes for drug repositioning via Enrichr. Querying the DSigDB database predicted several candidate compounds, including phorbol 12-myristate 13-acetate, indirubin (CHEMBL35349), tamibarotene, retinoic acid, pergolide, ropivacaine, lidocaine, aspirin, fenbuconazole, and methotrexate (Table 3). Search Tool for Interactions of Chemicals (STITCH) database analysis revealed interactions between retinoic acid and SPI1/ITGAM, between indirubin and SPI1, and between ropivacaine and ITGAM [29] (Fig 7A). The chemical structures of retinoic acid, indirubin, and ropivacaine are shown in Fig 7B–D. Molecular docking simulations confirmed stable binding conformations for these candidate drug–target pairs, with favorable binding energies (Fig 7E–H). These in silico results suggest testable hypotheses for modulating the shared pathogenic network.

**Table 3. AF and Hp gene-targeted drugs.**

| Term | P value | Adjusted P value | Combined Score | Genes |
|---|---|---|---|---|
| Phorbol 12-myristate 13-acetate (phorbol 12. 13.) | 3.72E-08 | 1.19E-05 | 1049.705997 | HCK; TYROBP; ITGAM; SPI1; NCF2; ITGB2 |
| CHEMBL35349 (indirubin) | 9.50E-07 | 1.52E-04 | 3120.361178 | CSF3R; SPI1; ITGB2 |
| Tamibarotene | 3.76E-06 | 4.02E-04 | 438.2370113 | HCK; TYROBP; ITGAM; FCER1G; ITGB2 |
| Retinoic acid | 7.22E-06 | 5.79E-04 | 394.727841 | HCK; FCGR3A; TYROBP; ITGAM; CSF3R; SPI1; FCER1G; NCF2; ITGB2 |
| pergolide | 1.13E-05 | 7.27E-04 | 485.1570488 | HCK; TYROBP; FCER1G; NCF2 |
| Ropivacaina (ropivacaine) | 2.04E-05 | 0.001091906 | 4494.847104 | ITGAM; ITGB2 |
| Lidocaine | 9.07E-05 | 0.004159024 | 1720.515946 | ITGAM; ITGB2 |
| aspirin | 1.12E-04 | 0.004259775 | 211.5027674 | HCK; CSF3R; FCER1G; ITGB2 |
| Fenbuconazole | 1.33E-04 | 0.004259775 | 1349.727435 | FCER1G; C5AR1 |
| methotrexate | 1.64E-04 | 0.004798672 | 182.995811 | TYROBP; CSF3R; FCER1G; C5AR1 |

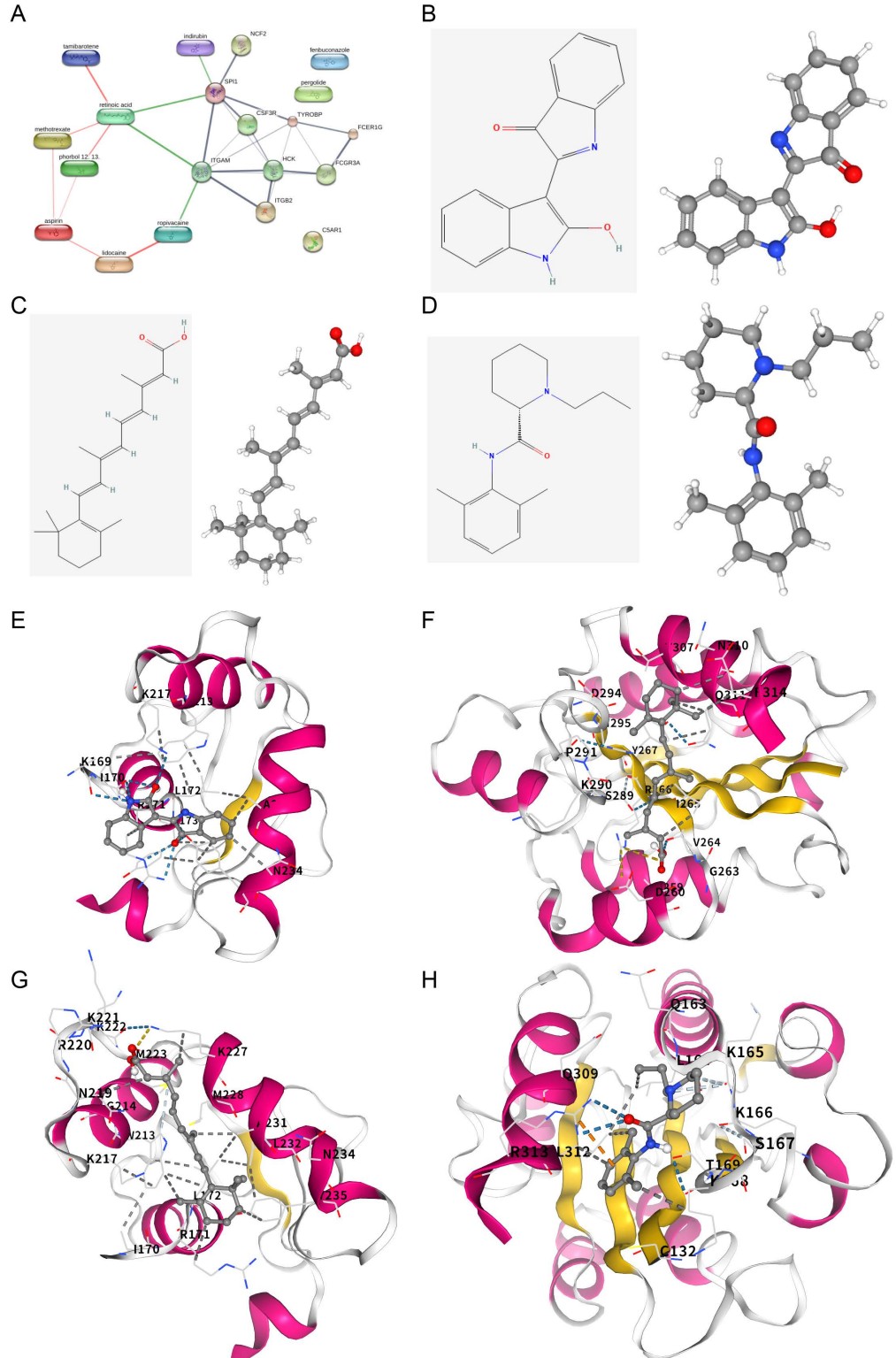

**Fig 7. Prediction and validation of potential therapeutic compounds.** (A) PPI network diagram of the 10 compounds and their predicted target hub genes, visualized using the STITCH database. (B–D) Chemical structures and 3D structures of indirubin (B), retinoic acid (C), and ropivacaine (D). (E–H) Molecular docking model showing the predicted binding conformations of indirubin and SPI1 (E), retinoic acid and ITGAM (F), retinoic acid and SPI1 (G), and ropivacaine and ITGAM (H).

## Discussion

Our integrative bioinformatics and ML study elucidates the shared molecular landscape between *H. pylori* infection and AF, with a focus on robust immune-inflammatory activation. This computational approach moves beyond reported epidemiological links [15,30–32] to define a precise, genetically based framework that could explain the clinical association. The identification of 77 unique genes at the intersection of both conditions, significantly enriched in pathways governing neutrophil chemotaxis, myeloid leukocyte activation, and leukocyte-mediated immunity, provides strong molecular evidence supporting chronic inflammation as a key mechanistic bridge. These findings align with and extend the current understanding that systemic inflammation induced by *H. pylori* [33–35] creates a proarrhythmic substrate, facilitating the electrophysiological and structural remodeling central to AF pathogenesis [36,37].

A pivotal discovery from our multialgorithm ML pipeline is the prominence of S100A8 as a shared diagnostic biomarker. S100A8, which typically functions as a heterodimer with S100A9, is a potent damage-associated molecular pattern (DAMP) protein crucial to innate immunity [38,39]. Its role appears to be context-dependent, linking local infection to systemic cardiac vulnerability. During *H. pylori* infection, S100A8/A9 contributes to gastric mucosal inflammation and host defense via nutritional immunity [40–43], potentially increasing its systemic expression. In the context of AF, S100A8/A9 promotes oxidative stress, cardiomyocyte dysfunction, and electrical remodeling [44–46]. Thus, we propose that S100A8 serves as a measurable pathogenic link: its systemic expression increase due to chronic gastric infection may directly exacerbate atrial inflammation and oxidative injury, thereby lowering the threshold for arrhythmogenesis. Our models suggest that quantifying S100A8 expression, potentially in conjunction with that of other key biomarkers such as FCER1G or ITGB2, could enhance risk stratification for AF in patients with *H. pylori* infection.

Beyond S100A8, the PPI network revealed a core set of hub genes—including TYROBP, ITGB2, SPI1, ITGAM, and C5AR1—that further underscore the centrality of leukocyte adhesion, signaling, and activation in this shared pathology. The enrichment of these hubs in pathways such as "neutrophil extracellular trap formation" and "myeloid leukocyte-mediated immunity" indicates that a sustained, coordinated innate immune response is a plausible unifying mechanism. This network analysis shifts the focus from a single biomarker to a dysfunctional immune module, offering a broader set of potential therapeutic targets for intervention.

The translational promise of our findings is highlighted by the drug repositioning analysis. Candidate compounds such as indirubin and ropivacaine, which are predicted to target hub genes such as SPI1 and ITGAM, merit investigation given their known anti-inflammatory properties in other contexts [47,48]. Particularly intriguing is the connection with retinoic acid, which has experimentally been shown to downregulate S100A8 expression [49]. This suggests a testable hypothesis: retinoic acid could attenuate the *H. pylori*-AF link by mitigating the S100A8-driven inflammatory cascade. The stable binding conformations of these candidates with their target proteins, validated by molecular docking, provide a structural rationale for further preclinical studies.

## Limitations and future perspectives

We acknowledge several limitations inherent to this in silico study. First, while we employed rigorous ComBat batch correction and independent validation cohorts, the analysis is based on heterogeneous public datasets, and the findings require confirmation in prospective, uniformly processed clinical cohorts. Second, our study demonstrated an association but could not establish causality between *H. pylori* infection, the identified gene signatures, and AF. Third, the diagnostic models, although performant, need validation in larger, multicenter studies to assess their real-world clinical utility. Finally, the predicted therapeutic candidates need thorough experimental validation in vitro and in vivo to confirm their efficacy and mechanism. Future research should employ Mendelian randomization to infer causality, utilize single-cell sequencing to pinpoint the specific immune cell populations driving these signatures, and conduct functional experiments to delineate the precise roles of hub genes such as S100A8 in atrial pathobiology.

## Conclusion

In conclusion, by integrating bioinformatics and machine learning, we constructed a shared immune–inflammatory pathway network linking *H. pylori* infection to AF, with S100A8 emerging as a central biomarker. The identified hub genes and the drug repositioning candidates, especially retinoic acid, provide a foundational framework for understanding the pathophysiology and for developing novel diagnostic and therapeutic strategies. This work translates epidemiological observations into a molecular hypothesis, offering concrete targets for future mechanistic and clinical investigations.

## Supporting information

**S1 Fig. Functional enrichment and pathway enrichment analysis of DEGs in *H. pylori* infection and AF.** Dot plot (A, D) and circular plot (B, E) displaying the results of the GO enrichment analysis of DEGs specific to *H. pylori* infection and AF. KEGG enrichment analysis results of DEGs specific to *H. pylori* infection (C) and AF (F).
(TIF)

**S2 Fig. Batch effect correction of AF and *H. pylori* datasets.**
(TIF)

**S1 Table. 77 union genes for comprehensive functional annotation and machine learning.**
(DOCX)

**S2 Table. AUC values for various models across cohorts in H. pylori infection.**
(DOCX)

**S3 Table. AUC values for various models across cohorts in AF.**
(DOCX)

**S4 Table. The AUC values of the top 5 genes incorporated in the optimal ML methods for H. pylori infection.**
(DOCX)

**S5 Table. The AUC values of the top 5 genes incorporated in the optimal ML methods for AF.**
(DOCX)

## Author contributions

**Conceptualization:** Wei Wang.

**Data curation:** Wei Wang.

**Formal analysis:** Wei Wang.

**Funding acquisition:** Aojian Deng.

**Investigation:** Wei Wang, Aojian Deng.

**Project administration:** Wei Wang.

**Resources:** Wei Wang, Aojian Deng.

**Software:** Aojian Deng.

**Supervision:** Aojian Deng.

**Validation:** Aojian Deng.

**Visualization:** Wei Wang, Aojian Deng.

**Writing – original draft:** Wei Wang, Aojian Deng.

**Writing – review & editing:** Wei Wang.

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
