## [Decision Letter · Decision Letter 0]

17 Dec 2025

Dear Dr. wang,

Thank you for submitting your manuscript to PLOS ONE. After careful consideration, we feel that it has merit but does not fully meet PLOS ONE’s publication criteria as it currently stands. Therefore, we invite you to submit a revised version of the manuscript that addresses the points raised during the review process.

We look forward to receiving your revised manuscript.

Kind regards,

Tomasz W. Kaminski

Academic Editor

PLOS One

**Journal Requirements:**

“None”

5. Please note that your Data Availability Statement is currently missing a direct link to access each database. If your manuscript is accepted for publication, you will be asked to provide these details on a very short timeline. We therefore suggest that you provide this information now, though we will not hold up the peer review process if you are unable.

7. We note you have included a table to which you do not refer in the text of your manuscript. Please ensure that you refer to Table 1 in your text; if accepted, production will need this reference to link the reader to the Table.

8. Please include captions for your Supporting Information files at the end of your manuscript, and update any in-text citations to match accordingly. Please see our Supporting Information guidelines for more information: http://journals.plos.org/plosone/s/supporting-information ..

**Additional Editor Comments:**

Dear Authors,

Thank you for your submission. The study addresses an interesting question, and the overall analytical approach is promising. However, substantial revisions are needed to improve methodological clarity, manuscript structure, figures, formatting and language quality before the work can be reconsidered.

We look forward to receiving your re-submission.

Best Regards,

Tomasz W Kaminski

Reviewers' comments:

Reviewer's Responses to Questions

**Comments to the Author**

1. Is the manuscript technically sound, and do the data support the conclusions?

Reviewer #1: Partly

Reviewer #2: Yes

2. Has the statistical analysis been performed appropriately and rigorously?

Reviewer #1: No

Reviewer #2: Yes

3. Have the authors made all data underlying the findings in their manuscript fully available?

Reviewer #1: Yes

Reviewer #2: Yes

4. Is the manuscript presented in an intelligible fashion and written in standard English?

Reviewer #1: No

Reviewer #2: Yes

Reviewer #1: This manuscript, in its current form, does not meet the minimum presentation, structure, and formatting standards expected of a PLOS ONE research article, independently of the scientific content. Even before considering the validity of the analyses, the paper does not look or read like a publication-ready journal article. This is a major concern and would normally justify rejection without peer review or a request for complete reformatting and resubmission.

1. Overall Structure Does Not Follow Journal Standards

While the manuscript nominally contains the standard sections (Abstract, Introduction, Methods, Results, Discussion), the structure is not consistently applied in a journal-ready way:

The Methods section reads like a rough technical checklist, not a properly written scientific methods narrative.

The Results section lacks clear subheadings and logical flow, making it difficult to follow the analytical progression.

The Discussion repeatedly re-states the Results instead of critically interpreting them, and frequently shifts into speculation without clear boundaries.

The manuscript currently reads more like a bioinformatics project report or preprint draft, not a polished research article.

2. Figures and Tables Are Not Presented at Journal Standard

The figures and tables exhibit multiple serious presentation problems:

Repeated typographical errors such as "Fiugre" instead of "Figure" throughout the document are unprofessional and unacceptable at submission stage.

Figure references appear as “Click here to access/download;Figure;Figure X.tif”, which is not appropriate for a manuscript text and suggests that the document was exported incorrectly.

The figures themselves are not described with sufficient interpretive captions — many captions merely restate what the plot type is, rather than what the figure shows scientifically.

Table formatting is inconsistent, and some entries appear cluttered, poorly aligned, and difficult to read.

At minimum, the authors must:

Correct all figure and table labeling errors,

Embed figures and tables properly in publication format,

Rewrite all figure legends to be self-contained and explanatory, not technical placeholders.

3. Language, Grammar, and Editorial Quality Are Substandard

While the general meaning of the text is understandable, the manuscript contains:

Numerous grammatical errors, awkward phrasing, and unnatural sentence construction,

Inconsistent spacing, punctuation, and formatting,

Repetitive wording and poorly structured paragraphs,

Informal or imprecise scientific phrasing.

PLOS ONE explicitly states that it will not copyedit accepted manuscripts, and the current language level is not acceptable for publication. The manuscript requires professional, full-scale English language editing, not light proofreading.

4. Visual and Logical Flow Is Disrupted

The overall visual and logical presentation is weak:

Figures, tables, and text are not well integrated.

The narrative jumps between bioinformatics steps without smooth transitions.

The manuscript lacks a clear storyline that would guide a reader from biological question → computational strategy → biological interpretation.

As it stands, the paper is difficult to read continuously as a coherent article.

5. Professional Presentation and Journal Readiness

Taken together:

Formatting problems,

Figure embedding errors,

Language quality,

Structural weaknesses,

mean that this manuscript does not resemble a finished journal article. It appears closer to a working draft or early internal report rather than a manuscript ready for peer-reviewed publication.

- Final Recommendation on Presentation Grounds Alone

Regardless of the scientific analyses, I believe this manuscript requires complete reformatting and professional language editing before it can be properly evaluated as a journal article.

At this stage, my recommendation based purely on presentation, structure, and formatting is:

Major Revision (borderline Reject / Resubmit as New Submission after full reformatting)

Reviewer #2: Overall, this manuscript is clearly organized and presents an interesting integrative analysis combining bioinformatics and machine learning to explore shared molecular features between H. pylori infection and atrial fibrillation.

To strengthen transparency and reproducibility, it would be helpful to add a concise dataset and preprocessing summary, including for each GEO dataset the platform, source, sample sizes per group, and the key preprocessing steps, as well as a clear description of how batch effects were addressed when combining discovery datasets.

The machine-learning component is promising, and a few additional reporting details would make the work easier to reproduce and interpret. Specifically, (i) how the discovery datasets were harmonized/combined prior to modeling, (ii) which hyperparameters were evaluated and the criterion used to choose the final settings, (iii) how the final gene panels were derived, and (iv) how the integrated models were implemented (stacking, voting, or sequential). If cross-validation was used within the discovery cohort during tuning or feature selection, please specify the CV design; if not, stating that explicitly would also help readers interpret the reported AUCs. Providing the final gene lists, key model settings, and any relevant code and parameters in supplementary material would further increase confidence in the robustness of the proposed markers.

The manuscript is generally intelligible and readable, and with a careful proofread it could be even stronger. I suggest revising minor typographical and formatting issues (like "Fiugre")and ensuring consistent use of abbreviations, spacing, and figure labels throughout the text.

**Do you want your identity to be public for this peer review?** For information about this choice, including consent withdrawal, please see our For information about this choice, including consent withdrawal, please see our Privacy Policy .

Reviewer #1: No

Reviewer #2: **Yes:** Yuchen ZhangYuchen Zhang

---

## [Author Response · Author response to Decision Letter 1]

12 Jan 2026

Manuscript Title: Integrative Bioinformatics and Machine Learning Identify Shared Molecular Mechanisms and Diagnostic Biomarkers Between Helicobacter pylori Infection and Atrial Fibrillation

We sincerely thank the editor and both reviewers for their valuable time and insightful comments, which have greatly helped us improve the quality and clarity of our manuscript. We have carefully considered each point and have substantially revised the manuscript accordingly. In direct response to your critique that the manuscript did not meet minimum journal standards, we undertook a comprehensive overhaul. We have systematically addressed every point raised by carefully re-examining, re-writing, and professionally refining the entire manuscript. Below, we provide a detailed point-by-point account of the actions taken.

Response to Reviewer #1:

Comment 1: On overall structure, language, grammar, and editorial quality.

Response: We agree completely with the reviewer's assessment that the original manuscript's language and structure were substandard and read more like a project report. To rectify this, we have not merely edited but have substantially re-written the manuscript to conform to the standards of a research article.

Global Language and Formatting Revision: The entire text has undergone rigorous professional language editing and proofreading. We have corrected grammatical errors, improved sentence fluency and clarity, eliminated awkward phrasing and repetition, and ensured consistent and formal scientific terminology throughout.

Restructured Methods Section: The "Materials and Methods" section has been completely re-organized and re-written. It now provides a coherent, logical narrative of our analytical workflow in dedicated subsections, moving decisively away from the initial checklist format to offer clear, reproducible descriptions.

Re-organized and Re-written Results Section: We have introduced clear, descriptive subheadings that reflect the analytical progression The text has been carefully re-crafted to ensure a logical flow, explicitly connecting each analytical step to the next and guiding the reader clearly through the sequence of discoveries.

Expanded and Deepened Discussion Section: We have significantly expanded the "Discussion" to move decisively beyond re-stating results. It now provides a critical interpretation of our findings within the broader context of existing literature, elaborates in depth on the biological and clinical implications (e.g., the dual contextual role of S100A8), and clearly differentiates between evidence-based conclusions and speculative points for future inquiry. A dedicated "Limitations and Future Perspectives" subsection has been added to frame the study appropriately.

Correction of Editorial Artifacts: Regarding the specific typographical error ("Fiugre") and improper figure references noted, we have meticulously proofread our source documents. We believe these may have been introduced during the initial file export or submission process. We will exercise utmost care during re-submission to ensure all text, labels, and citations are error-free.

Comment 2: On the presentation of figures/tables and the overall logical/visual flow.

Response: We thank the reviewer for this critical feedback on presentation, which we have addressed comprehensively.

Enhanced Figures and Tables: All figures have been re-created and formatted strictly according to typical journal requirements concerning font consistency, size, resolution, and visual clarity. The legends have been entirely re-written to be self-contained and explanatory, detailing what each panel shows and summarizing the key scientific finding it conveys.

Improved Logical Narrative and Cohesion: To address the disjointed flow, we have strengthened the connective tissue of the manuscript. This includes refining the narrative arc from the Introduction through to the Discussion and incorporating more transitional phrasing between sections and paragraphs. This ensures a smoother, more coherent reading experience that logically connects the biological question, the computational strategy, and the final interpretation.

We are profoundly grateful for the reviewer's rigorous assessment. The critical feedback provided was essential for us to elevate the quality of our work. We believe the extensive revisions detailed above have thoroughly addressed all the concerns raised and have transformed the manuscript into a polished, coherent, and publication-ready research article. We hope the revised version now meets the expected standards of your journal.

Response to Reviewer #2:

We sincerely thank the reviewer for the positive assessment and exceptionally constructive suggestions, which have been instrumental in guiding our comprehensive revision. As noted in our response to Reviewer #1, we have re-written the manuscript in its entirety to address overarching presentation issues. Your specific comments were central to shaping the new, detailed methodological reporting within this re-written framework, greatly enhancing the work's transparency and reproducibility.

Comment 1: Strengthening transparency in dataset description and preprocessing.

Response: We have added a new subsection, and a Supplementary Table 1 summarizing each dataset (accession, platform, sample sizes). We now clearly state that ComBat batch correction was applied to the discovery cohorts only, while validation cohorts were processed independently.

Comment 2: Clarifying the machine learning workflow.

Response: We have expanded the methods subsection “Machine Learning-Based Diagnostic Model Construction and Validation” to detail the process:

Hyperparameter tuning was performed using 10-fold cross-validation repeated 5 times within the discovery cohort.

Final gene panels were derived by intersecting top features from multiple algorithms.

The integrated models were implemented via a two-step approach (base RF + meta-classifier).

All reported AUROC values are from the independent validation cohorts, ensuring an unbiased performance estimate.

Comment 3: Improving language and formatting consistency.

Response: The manuscript has undergone thorough proofreading. All typographical errors have been corrected, and formatting (abbreviations, spacing, figure labels) has been standardized throughout.

We are grateful for your insightful comments, which have strengthened the manuscript.

---

## [Decision Letter · Decision Letter 1]

27 Jan 2026

Dear Dr. wang,

Thank you for submitting your manuscript to PLOS ONE. After careful consideration, we feel that it has merit but does not fully meet PLOS ONE’s publication criteria as it currently stands. Therefore, we invite you to submit a revised version of the manuscript that addresses the points raised during the review process.

We look forward to receiving your revised manuscript.

Kind regards,

Tomasz W. Kaminski

Academic Editor

PLOS One

**Journal Requirements:**

**Additional Editor Comments:**

Dear Authors,

Thank you for your revised submission and for the careful responses provided in the previous round. One reviewer finds that the manuscript has adequately addressed all earlier comments and considers the study technically sound and suitable for publication.

To further strengthen the manuscript, I encourage you to focus the revision on improving the clarity and structure with which the key results are presented and interpreted, as suggested Reviewer. In particular, summarizing the main findings in a more structured manner and expanding the analytical depth of the Discussion - especially the limitations - would help readers more clearly assess the scope, robustness, and implications of the work.

These revisions are intended to enhance transparency and balance rather than to request additional analyses. Please feel free to reach out if any points in the reviews would benefit from clarification.

Kind regards,

Tomasz W Kaminski

Reviewers' comments:

Reviewer's Responses to Questions

**Comments to the Author**

Reviewer #1: All comments have been addressed

Reviewer #2: All comments have been addressed

2. Is the manuscript technically sound, and do the data support the conclusions?

Reviewer #1: Partly

Reviewer #2: Yes

3. Has the statistical analysis been performed appropriately and rigorously?

Reviewer #1: Yes

Reviewer #2: Yes

4. Have the authors made all data underlying the findings in their manuscript fully available?

Reviewer #1: Yes

Reviewer #2: Yes

5. Is the manuscript presented in an intelligible fashion and written in standard English?

Reviewer #1: Yes

Reviewer #2: Yes

Reviewer #1: The manuscript lacks clear, publication-quality figures and tables that adequately support the results. At present, the results are presented largely in narrative form, with minimal quantitative visualization. Key findings (e.g., diagnostic model performance, hub gene importance, validation outcomes) are not summarized in structured tables or interpretable figures.

The absence of well-designed figures and summary tables significantly limits the evidential value of the manuscript and makes it difficult for readers to independently assess the results. The authors should include properly labelled, publication-standard figures and tables that directly correspond to each major result and hypothesis.

While the Discussion is extensive, it remains largely narrative and would benefit from deeper analytical expansion rather than additional speculative content. In particular, the limitations section should be substantially expanded to address dataset heterogeneity, model stability, lack of causal inference, and the non-predictive nature of molecular docking. Strengthening these sections would significantly improve the scientific rigor and interpretive balance of the manuscript.

Reviewer #2: The authors have adequately addressed all comments raised in the previous review. The revised manuscript is clearly written, technically sound, and presents a coherent and well-documented analytical workflow. The data support the conclusions, and the study meets the standards for publication.

**Do you want your identity to be public for this peer review?** For information about this choice, including consent withdrawal, please see our For information about this choice, including consent withdrawal, please see our Privacy Policy .

Reviewer #1: No

Reviewer #2: **Yes:** yuchen zhangyuchen zhang

---

## [Author Response · Author response to Decision Letter 2]

12 Mar 2026

Response to Reviewer #1：

Dear Reviewer,

We sincerely thank you for re-reviewing our manuscript and for providing such valuable feedback. Your professional insights have been instrumental in further improving the quality of our work. In this second round of revision, we have carefully considered all the issues raised and have made corresponding improvements to the manuscript. Below is our point-by-point response to your specific comments.

Response to Reviewer #1:

Comment 1: The manuscript lacks clear, publication-ready data and tables to adequately support the results. Key findings are not summarized in structured tables or interpretable figures, which limits the evidentiary value of the manuscript.

Response: We fully agree with your assessment that clear data visualization and structured tables are fundamental to supporting scientific conclusions. To address this, we have comprehensively supplemented and optimized the figures and tables in our manuscript: Addition of a study flowchart (now Figure 1): We have included a comprehensive flowchart that systematically illustrates the entire research workflow, from data acquisition and preprocessing, through differential expression analysis, WGCNA, machine learning, and hub gene screening, to drug prediction. This visual summary helps readers quickly grasp the overall study design. Tabular presentation of machine learning features and performance (new S2 Table and S3 Table): The pivotal diagnostic genes for H. pylori and AF (previously shown only in Figure 4A and 4F) are now summarized, which also lists each gene's SHAP value and ranking, enabling quantitative assessment of feature importance. The diagnostic performance metrics (AUROC) of the final models are now organized in S4 Table and S5 Table, with results clearly labeled for both the discovery cohort (internal cross-validation) and independent validation cohorts. This makes the model validation process more transparent and credible. Visualization of batch effect correction (new Fig S2): To address concerns regarding dataset heterogeneity, we have added PCA plots and boxplots for both AF and H. pylori datasets before and after ComBat correction. This figure intuitively demonstrates that batch effects were effectively eliminated and samples from different datasets were well-integrated, providing visual justification for subsequent combined analyses. Complete list of core genes (new S1 Table): We have provided the full list of 77 union genes (including gene symbols, full names, and expression patterns) as a supplementary table, facilitating direct access and analysis by readers. All figures and tables have been prepared according to journal formatting guidelines (appropriate resolution, font size, layout), and the legends have been rewritten to be self-explanatory. We believe these additions substantially enhance the readability and evidentiary value of our results.

Comment 2: The Discussion remains largely narrative. It would benefit from deeper analysis rather than speculative expansion. In particular, the Limitations section should be substantially expanded to address dataset heterogeneity, model stability, lack of causal inference, and the non-predictive nature of molecular docking.

Response: We thank you for your insightful comments on the Discussion. In our revision, we have placed particular emphasis on enhancing the depth and critical analysis of this section: Strengthened interpretation grounded in results: The newly added figures and tables (Figure 1 flowchart, Tables 2 and 3, Supplementary Figure 2, etc.) provide a more robust data foundation for the core arguments in the Discussion. We have closely integrated these new results into the Discussion, offering in-depth interpretations of key findings (such as the dual role of S100A8 and the immune network status of hub genes) while avoiding unsubstantiated speculation. Clear distinction between conclusions and hypotheses: We have carefully reviewed all discussion statements to ensure that every inference is grounded in our results or published literature. Hypothetical points (such as the therapeutic potential of candidate drugs) are expressed with appropriate caution, explicitly noting the need for future validation. Expanded limitations discussion: Following your suggestion, we have further emphasized the following aspects in the "Limitations and Future Perspectives" subsection: Dataset heterogeneity: Although ComBat correction was applied, the original data originated from different platforms; future validation using prospective, uniformly processed cohorts is still warranted. Model stability: While the models performed well in independent validation, broader external validation in multi-center studies would further strengthen their generalizability. Lack of causal inference: This study is associational and cannot establish causality; future studies employing Mendelian randomization or longitudinal designs are needed to explore causal relationships. Predictive nature of molecular docking: The docking results are computational predictions; their biological activity, specificity, and safety require experimental validation.

These additions have made the Limitations section more comprehensive and balanced, reflecting our clear awareness of the scientific boundaries of our conclusions.

We once again thank you for taking the time to review our manuscript. Your rigorous and detailed feedback has pushed us to pursue higher standards. We would also like to express our sincere gratitude to Reviewer #2 for their positive evaluation and recommendation for publication. The professional insights from both reviewers have jointly contributed to the improvement of this study, and we are deeply appreciative.

We hope that the revised manuscript now meets your expectations.

Sincerely.

---

## [Editor Report · Decision Letter 2]

15 Mar 2026

Integrative Bioinformatics and Machine Learning Identify Shared Molecular Mechanisms and Diagnostic Biomarkers Between Helicobacter pylori Infection and Atrial Fibrillation

PONE-D-25-59195R2

Dear Dr. wang,

We’re pleased to inform you that your manuscript has been judged scientifically suitable for publication and will be formally accepted for publication once it meets all outstanding technical requirements.

Kind regards,

Tomasz W. Kaminski

Academic Editor

PLOS One

---

## [Editor Report · Acceptance letter]

PONE-D-25-59195R2

PLOS One

Dear Dr. Wang,

I'm pleased to inform you that your manuscript has been deemed suitable for publication in PLOS One. Congratulations! Your manuscript is now being handed over to our production team.

Kind regards,

on behalf of

Dr. Tomasz W. Kaminski

Academic Editor

PLOS One